# Improved Lead Sensing Using a Solid-Contact Ion-Selective Electrode with Polymeric Membrane Modified with Carbon Nanofibers and Ionic Liquid Nanocomposite

**DOI:** 10.3390/ma16031003

**Published:** 2023-01-21

**Authors:** Cecylia Wardak, Klaudia Morawska, Beata Paczosa-Bator, Malgorzata Grabarczyk

**Affiliations:** 1Department of Analytical Chemistry, Institute of Chemical Sciences, Faculty of Chemistry, Maria Curie-Sklodowska University, Maria Curie-Sklodowska Sq. 3, 20-031 Lublin, Poland; 2Faculty of Materials Science and Ceramics, AGH University of Science and Technology, Mickiewicza 30, 30-059 Krakow, Poland

**Keywords:** lead ion-selective electrode, all-solid-state ion-selective electrode, solid contact, carbon nanofibers and ionic liquid nanocomposite, lead determination, potentiometry

## Abstract

A new solid-contact ion-selective electrode (ISE) sensitive to lead (II) ions, obtained by modifying a polymer membrane with a nanocomposite of carbon nanofibers and an ionic liquid 1-hexyl-3-methylimidazolium hexafluorophosphate, is presented. Electrodes with a different amount of nanocomposite in the membrane (0–9% *w/w*), in which a platinum wire or a glassy carbon electrode was used as an internal electrode, were tested. Potentiometric and electrochemical impedance spectroscopy measurements were carried out to determine the effect of the ion-sensitive membrane modification on the analytical and electrical parameters of the ion-selective electrode. It was found that the addition of the nanocomposite causes beneficial changes in the properties of the membrane, i.e., a decrease in resistance and an increase in capacitance and hydrophobicity. As a result, the electrodes with the modified membrane were characterized by a lower limit of detection, a wider measuring range and better selectivity compared to the unmodified electrode. Moreover, a significant improvement in the stability and reversibility of the electrode potential was observed, and additionally, they were resistant to changes in the redox potential of the sample. The best parameters were shown by the electrode obtained with the use of a platinum wire as the inner electrode with a membrane containing 6% of the nanocomposite. The electrode exhibited a Nernstian response to lead ions over a wide concentration range, 1.0 × 10^−8^–1.0 × 10^−2^ mol L^−1^, with a slope of 31.5 mV/decade and detection limit of 6.0 × 10^−9^ mol L^−1^. In addition, the proposed sensor showed very good long term stability and worked properly 4 months after its preparation without essential changes in the E^0^ or slope values. It was used to analyze a real sample and correct results of lead content determination were obtained.

## 1. Introduction

The concentration of lead in the world around us continues to increase dramatically due to its extensive use; for example, in industry, for the production of lead-acid batteries, in agriculture, or in medicine as a component of radiation protection devices. Lead has no positive effects on the functioning of the human body in any way. It has been classified as a carcinogen, a mutagen that causes birth defects, and an agent that damages internal organs [1]. Lead is a neurotoxin that accumulates in the bones, body fluids and organs such as the liver, brain and kidneys after being assimilated by the body [2,3]. 

In view of the above, it is of high importance to develop better methods for determining lead concentration in samples with good precision, repeatability and a low limit of detection. With the help of analytical techniques such as spectroscopy and chromatography, or electrochemical methods such as voltammetry or potentiometry, we are also able to determine a specific element, such as lead, that is present in samples even in small amounts [4,5]. 

In electrochemical techniques, there is growing interest in potentiometry, where ion-selective electrodes (ISEs), mainly solid-contact ISEs, are used for measurements [6,7]. It is because, with their help, much better values of parameters such as potential stability and reversibility and a lower limit of detection are achieved. In addition, SC-ISEs have a number of advantages such as easy and fast measurement, low costs of production, quite high robustness, as well as the possibility of miniaturization [8,9]. This is certainly connected with the elimination of the inner filling solution, but unfortunately, as a consequence, disadvantages such as weaker stability and reproducibility of potential or increased resistance are also induced and observed [10]. The most common solution to get rid of these defects is to use an electroactive material known as a solid contact; for example, conducting polymers, carbon-based nanomaterials or metal oxide nanoparticles that will ensure a better charge transfer between the ion-sensitive membrane and the inner electrode, which in turn will improve the stability and reproducibility of the electrode potential [9,11]. This additional electroactive material can be used in two ways: as an intermediate layer between the ion-sensitive membrane and the inner electrode, or as a component of the ion-sensitive membrane [12,13].

Carbon-based nanomaterials are great to use as a solid contact in ion-selective electrodes [14,15]. They are used all over the world as a co-electrode material [16,17]. There is growing interest in them due to their optical, mechanical, but especially electrical properties [18]. Their quite significant advantage is their almost ideal electrical conductivity, which is very important in ion-selective electrodes, because this property facilitates the transport of a charge between the ion-selective membrane and electrical conductor and keeps the potential stabilized. It is also worth mentioning that they are chemically passive and have a large window of potential, as well as a low resistance which makes them a brilliant solid electrode contact [19,20,21,22]. 

In recent years, composite materials based on carbon nanomaterials, mainly carbon nanotubes [23,24] and carbon nanofibers [25,26], have been successfully used in solid-contact electrodes. These materials combine the desired properties of ingredients that often exhibit synergistic properties. Thanks to this, the use of a nanocomposite is more advantageous than a single component. First of all, the nanocomposite performs the basic function as an ion-electron transducer, and additionally, other benefits are obtained. 

Michalska’s group proposed a nanocomposite of multiwalled carbon nanotubes and poly(3-octylthiophene-2,5-diyl) (POT), in which POT was used as a dispersing agent for the carbon nanotubes. In addition, the immobilization of POT on carbon nanostructures prevents the unwanted transfer of this polymer to the membrane [23]. Paczosa et al. studied the properties of various composites obtained from ruthenium dioxide and carbon nanomaterials (graphene, multiwalled carbon nanotubes and carbon black) for their use as solid contacts in ISEs. They found that the addition of ruthenium dioxide caused an increase in the capacitance value for all tested carbon nanomaterials which positively affected the potentiometric response of the ion-selective electrodes containing the nanocomposite as an intermediate layer [27]. We observed a similar effect in our research in relation to the composite obtained from multi-walled carbon nanotubes and polyaniline nanofibers. This nanocomposite showed better electrical properties than its individual components (lower resistance and higher capacitance) and effectively acted as solid contact in chloride ion-selective electrodes [28]. Searching for new electroactive materials to improve the parameters of nitrate ISEs, we discovered a nanocomposite of multiwalled carbon nanotubes (MWCNTs) and ionic liquid-trihexyltetradecylphosphonium chloride (THTDPCl) [24]. In this case, the nanocomposite was used as a membrane component. The use of MMWCNTs in the form of a composite with an ionic liquid allows us to obtain a homogeneous membrane without the need for additional dispersants. It was shown that the use of the MWCNTs-IL nanocomposite as a membrane component simplifies the preparation of the electrode and at the same time improves its parameters to a greater extent than the use of MWCNTs as an intermediate layer [24].

In this paper, a new nanocomposite made of carbon nanofibers and an ionic liquid, 1-hexyl-3-methylimidazolium hexafluorophosphate (HMImPF_6_), is studied as a solid-contact material in a lead sensitive ion-selective electrode. 

Ionic liquids are organic chemical compounds composed entirely of ions, which are usually liquids at room temperature. These compounds have many interesting properties (low volatility, high ion conductivity, wide electrochemical window, thermal stability and flame retardancy) that allow their wide practical application. Among other things, ionic liquids are effectively used in electrochemistry [29,30] as electrolytes or additives to electrolytes in barriers [31], for the synthesis of electrode material [32] or in sensors as an electrode modifier [33,34].

HMImPF_6_ is a water-immiscible hydrophobic ionic liquid which possesses good extraction properties and was successfully used as an extraction solvent for lead preconcentration [35,36]. Is was also shown that this compound effectively acted as a lipophilic ionic component of lead sensitive ion-selective membranes [37]. HMImPF_6_ has an imidazolium ring with π electrons in its molecule, which allows it to interact with the π surface of carbon nanofibers. As a result of this interaction, the ionic liquid molecules surround the carbon nanofibers and cause steric and electrochemical stabilization of the resulting composite, which is easily dispersed in the membrane cocktail. It was also shown that the interaction of an imidazolium-based ionic liquid with the surface of carbon nanofibers improved their electrocatalytic properties [38] and increased electric capacitance [39]. Taking into account the valuable properties of these two nanocomposite components (hydrophobic properties, good electrical conductivity and mechanical resistance) and the benefits of their interaction, it seems that the nanocomposite obtained from them is an ideal/a perfect candidate for the modification of a membrane sensitive to lead ions, allowing us to obtain an ion-selective electrode with good analytical parameters that is easy to prepare and use. To the best of our knowledge, such a composite has not yet been studied as an electroactive material in ISEs.

## 2. Materials and Methods

### 2.1. Reagents

Lead ionophore IV-tert-butylcalix[4]arene-tetrakis(N,N-dimethylthioacetamide), 2-nitrophenyl octyl ether (NPOE), bis(1-butylpentyl) adipate (BBPA) and the ionic liquid 1-hexyl-3-methylimidazolium hexafluorophosphate (HMImPF_6_) were purchased from Fluka. Low-molecular-weight poly(vinyl chloride) (PVC) and carbon nanofibers (CNFs) (>99.9%) with a 100 nm diameter and a 20–200 µm length were obtained from Sigma Aldrich. Lead nitrate (pure pro analysis), other nitrate salts used in the interference study and other reagents were purchased from Fluka. All aqueous solutions were prepared using freshly deionized water.

### 2.2. Apparatus

The potentiometric measurements were performed using a cell consisting of the tested ion-selective electrodes and a silver/silver chloride reference electrode with a double junction system (Metrohm 6.0750.100), and a 16-channel data acquisition system (Lawson Labs. Inc., Malvern, PA, USA) connected to a computer with appropriate software was used for data collection. The electromotive force (EMF) measurements were made at room temperature (25 °C ± 1) in mixed solutions using a magnetic stirrer.

Electrochemical impedance spectroscopy measurements were carried out on a 3-electrode system in which the tested ion-selective electrode was the working electrode, Ag/AgCl (Metrohm 6.0733.100) was the reference electrode and platinum wire was the auxiliary electrode. All measurements were made in a Pb(NO_3_)_2_ solution with a concentration of 10^−2^ mol L^−1^. The impedance spectra were recorded in the frequency range 0.01–100 kHz at an open circuit potential with an amplitude of 10 mV at room temperature (25 °C ± 1) using an electrochemical analyzer AUTOLAB (Eco Chemie, Netherlands) with NOVA 2.1 software.

### 2.3. Preparation of Ion-Selective Electrode

Two types of internal electrodes were used to construct the ion-selective electrodes: platinum wire (Pt) and glassy carbon electrodes (GCE). Before they were covered with a membrane mixture, they were properly prepared. The Pt electrode was polished with fine-grained abrasive paper, then rinsed with water and degreased with acetone. The GCE was cleaned with fine-grained abrasive paper, then polished with alumina powder (0.3 µm size), wetted with distilled water, rinsed thoroughly and finally degreased with acetone.

The nanocomposite (NC) of the carbon nanofibers and ionic liquid was prepared by mixing its components, CNFs and HMImPF_6_, in a weight ratio of 1:5 and homogenizing the mixture in an ultrasonic bath for 30 min. The composite prepared in this way was used as a component of the ion-selective membrane (ISM) mixtures. Their composition is presented in Table 1.

The chemical formulas of the compounds used to prepare the membranes are shown in Figure 1.

After weighing the individual components, the membrane mixtures were thoroughly mixed and then deaerated using a vacuum pump. Then, the mixture was applied to the surface of the inner electrode (Pt or GCE) and gelled at 90 °C for 5 min. On the next day, the electrodes were conditioned in a Pb(NO_3_)_2_ solution with a concentration of 1 × 10^−3^ mol L^−1^ for 30 minutes, and then, before each measurement, in a Pb(NO_3_)_2_ solution with a concentration of 1 × 10^−5^ mol L^−1^ for 30 minutes. All conditioning procedures were performed at room temperature. Between measurements, the electrodes were stored in the air. The construction of the prepared electrodes is shown in Figure 2.

## 3. Results and Discussion

### 3.1. Potentiometric Response

To study the effect of the addition of the CNF and HMImPF_6_ nanocomposite to the membrane phase, the calibration curves of the prepared electrodes were determined. The measurements were completed in the Pb(NO_3_)_2_ solutions in the concentration range of 1 × 10^−9^–1 × 10^−2^ mol L^−1^. From the response curves of the tested electrodes shown in Figure 3, the range and slope of the rectilinear section of the characteristic, as well as the limit of detection were determined.

The response time was determined from the measurements of the potential as a function of time. It was assumed that the time that elapses from the moment when the Pb^2+^ concentration in the sample solution was changed to the first instance when the ∆E/∆t ≤ 0.2 mV/min. The analytical parameters of the tested electrodes are summarized in Table 2.

Based on the obtained results, it was found that the electrodes containing NC in the membrane compared to the coated wire/coated disc electrode without the nanocomposite were characterized by a reduced detection limit and a wider measuring range. This effect was dependent on both the membrane composition and the type of internal electrode. As the quantity of the NC in the membrane increased, a lower detection limit and a wider measuring range of the electrodes were observed. However, for an NC content of 9% in the membrane, two linearity ranges with different slopes were observed; a Nernstian one in the range of higher concentrations and a supernernstian one in the range of lower concentrations. Improved lead sensing by the electrodes with membrane modified by nanocomposite can be explained by the increased extraction capacity of the membrane in relation to lead. The increase in the nanocomposite content in the membrane increases the extractability of lead ions to the membrane phase. In the case of the Pt/ISM (9%NC) and GCE/ISM (9%NC) 3 electrodes, this effect is the strongest, causing a supernernstian slope of the electrode’s response curve in the low concentration range. Comparing electrodes with the same ion-selective membrane, but based on different internal electrodes, it can be concluded that the modification of the membrane with a nanocomposite was more effective in the case of the electrodes constructed on the basis of a platinum electrode. The insertion of NC to the membrane also had a positive effect on the response time of the electrodes, which was clearly shorter for the modified electrodes. The response time of the electrodes depends on the concentration of lead in the sample solutions and is shorter in solutions with a higher lead content/quantity. The response time values given in Table 2 correspond to solutions with a Pb^2+^ ion content of ≥ 1 × 10^−6^ mol L^−1^. The best performance of all tested sensors was shown by the Pt/ISM (6%NC) electrode, which was characterized by a wider measuring range and a lower detection limit by almost two orders of magnitude compared to the unmodified electrode. This Pt/ISM (6%NC) electrode was selected for further research to determine the remaining analytical parameters and to check its analytical usefulness. For comparison, the unmodified electrode Pt/ISM (0) was also tested in parallel.

### 3.2. Potential Stability, Reversibility and Reproducibility

Potential stability and reversibility of an ion-selective electrode are important parameters from the point of view of its practical application. Insufficiently good values of these parameters are the cause of measurement difficulties and a source of errors in the determination of results.

Short-term potential stability was studied by determining the potential change during continuous measurements of the potential over time in a 1 × 10^−2^ mol L^−1^ solution of lead ions for 3 h. The exemplary record of potential changes recorded during this experiment for the electrode with the 6% nanocomposite in the membrane and for the unmodified electrode is shown in Figure 4.

As can be seen, the Pt/ISM (6%NC) electrode with the nanocomposite-modified membrane is characterized by much more stable potential than the unmodified Pt/ISM (0) electrode. The determined potential drift was 1.75 and 142 µV/min for the Pt/ISM (6%NC) and Pt/ISM (0) electrodes, respectively.

The potential reversibility of the tested electrodes was determined by measuring their potential alternately in Pb(NO_3_)_2_ solutions with a concentration of 1 × 10^−4^ and 1 × 10^−5^ mol L^−1^, four times in each solution. The average potential values determined for the Pt/ISM (6%NC) electrode were 160.14 ± 0.11 and 137.8 ± 0.22 mV for the 1 × 10^−4^ and 1 × 10^−5^ mol L^−1^ Pb(NO_3_)_2_ solutions, respectively. The analogous values determined for the Pt/ISM (0) electrode are 184.3 ± 3.3 and 212.0 ± 2.4 mV. The comparison of the standard deviation of the potential for both electrodes clearly indicates the beneficial effect of the presence of the proposed nanocomposite in the membrane on the reversibility of the electrode potential. The improvement in the stability and reversibility of the potential is related to the modification of the membrane and the change in its properties, i.e., the reduction in its resistance and the acceleration of the charge transfer process at the interface of the ion-sensitive membrane with the inner electrode. This was confirmed by the EIS measurements described below.

The long-term stability of the potential is related to the lifetime of the electrode and determines the invariability of electrode indications for a solution of the same concentration during measurements over a long period of time. The long-term stability of the tested electrodes was estimated by the results obtained during systematic calibrations in freshly prepared solutions performed twice a week for a period of 4 months.

Changes in the E^0^ value and the slope of the characteristic were taken into account. The E^0^ potential values were determined from the calibration curves by extrapolation of its linear section to the pa_Pb (II)_ = 0. The results of these measurements obtained for three identical sensors of each type are presented in Figure 5, where it can be seen that the modified electrode was also characterized by a very good long-term stability and reproducibility.

It is known that in the case of electrodes with a polymer membrane, the change in the value of E^0^ over time, as well as the decrease in electrode sensitivity are related, among others, to the change in the composition of the membrane caused by the leaching of active ingredients from it during the use of the electrode. Based on the data presented in Figure 5, it can be concluded that the modification of the membrane with the addition of a nanocomposite increases its hydrophobicity, which effectively reduces leaching of active ingredients from the membrane phase into the water phase. Thanks to this, the obtained electrode does not require frequent control calibrations during its practical analytical use for determinations.

### 3.3. Selectivity

The effect of membrane modification on the electrode’s selectivity was assessed by determining the values of its selectivity coefficients. The method of separate solutions with fixed activity was used. The determined values of selectivity coefficients are listed in Table 3. From the analysis of the data contained in this table, it can be concluded that the electrode with a membrane containing the NC showed more favorable selectivity coefficient values compared to the electrode with a membrane of conventional composition presented in this paper as well as that described in the literature with the same ionophore [40]. This effect is related to the modification of the membrane composition with the use of the NC. The membrane of the Pt/ISM (0) electrode contains KTpClPB, which exhibits ion-exchange properties and increases the affinity of the membrane for all cations. The ionic liquid contained in the NC added to the membrane of the Pt/ISM (6%NC) electrode does not exhibit ion exchange properties, but it provides ions and increases the ionic strength of the membrane. As a result, ions with high affinity to the ionophore, in this case Pb^2+^ ions, are more easily transferred to the membrane phase, which in turn improves the selectivity of the electrode.

### 3.4. Optimal pH Range

The optimal pH range in which the electrode potential is stable and does not depend on the pH of the sample was determined. For this purpose, the electrode potential was measured in a 1.0 × 10^−4^ mol L^−1^ Pb(NO_3_)_2_ solution in the pH range of 2.0–8.0. The pH was adjusted using HNO_3_ or NaOH solutions. The dependence of the electrode potential on the pH of the sample solution for the Pt/ISM (0) and Pt/ISM (6%NC) electrodes is shown in Figure 6, where it can be seen that the NC-based electrode exhibited a wider pH range (3.1–7.6) than the unmodified electrode (4.0–7.4) and was more resistant to the interfering effect of H^+^ ions.

### 3.5. Redox Sensitivity

Due to the fact that carbon materials show redox sensitivity, the influence of the redox potential of the sample on the potential of the tested lead electrodes was checked. For this purpose, measurements of the electrode potential in solutions containing a constant concentration of Pb^2+^ ions (1.0 × 10^−3^ mol L^−1^), as well as Fe^2+^ and Fe^3+^ ions, were carried out, for which the total concentration was constant (1.0 × 10^−3^ mol L^−1^), where [Fe^3+^]/[Fe^2+^] was equal to 0.1, 0.2, 1.0, 5.0 and 10.0, respectively. The results of these measurements are shown in Figure 7 where it can be seen that the potential of tested electrodes does not depend on the change in the potential redox of the sample solution. Small changes in the potential of the unmodified electrode are due to potential instability rather than redox sensitivity.

### 3.6. Electrochemical Impedance Spectroscopy Measurements

In order to confirm the hypothesis that the presence of a nanocomposite in the membrane phase improves its electric properties, EIS measurements were conducted. Impedance spectra were recorded at an open circuit potential in the frequency range 0.01–100 kHz with an amplitude of 10 mV. The obtained results are shown in Figure 8, where plots with a shape typical for ISEs with a polymeric membrane can be seen. They consist of two parts, a semicircle in the high- and medium-frequency range, related to the bulk resistance of the membrane and its geometric capacitance, and a line in the low-frequency range attributed to the charge transfer resistance and the double layer capacitance between the membrane and the electronic conductor [41]. Although all impedance spectra have the same shape, they differ drastically in size, which indicates differences in the electrical properties of the electrodes.

The values of the electrical parameters of the electrodes (Table 4) were determined by fitting the data obtained in the EIS measurements to the equivalent circuit shown in the insert of Figure 8.

The data presented in Table 4 clearly indicate that the modification of the membrane with the nanocomposite results in a drastic reduction in the membrane resistance and the charge transfer resistance. This effect depends on the content of the nanocomposite in the membrane and is stronger for the electrode with a membrane with a higher content of the NC. The resistance of the unmodified membrane was 762 kΩ, which decreased almost five times after the introduction of the nanocomposite to a value of 178 kΩ for the Pt/ISM (3%NC) electrode and more than sixteen times to a value of 46.5 kΩ for the Pt/ISM (6%NC) electrode. The change in charge transfer resistance was even greater. The R_ct_ value of 28.6 kΩ determined for the Pt/ISM (6%NC) electrode was drastically lower compared to the R_ct_ value obtained for the Pt/ISM (0) electrode, which was 9832 kΩ. A significant reduction in the charge transfer resistance was also observed in the case of the Pt/ISM (3%NC) electrode, for which the R_ct_ value was 196 kΩ. However, due to the lower content of the NC in the membrane, this effect was weaker than for the Pt/ISM (6%NC) electrode. The improvement in the electrical parameters of the electrodes is the effect of modifying the composition of the membrane. The nanocomposite added to the membrane is characterized by very good electrical conductivity and effectively reduces the resistance of the membrane. The dispersion of NC throughout the entire volume of the membrane makes the contact surface of the membrane phase and NC very large. This significantly facilitates the diffusion processes and charge transport in the membrane phase and at the membrane /inner electrode interface and, as a result, effectively improves the stability and reversibility of the electrode potential.

### 3.7. Analytical Application of Proposed Electrode for Lead Determination in Real Samples

In order to show the analytical usefulness of the developed Pt/ISM (6%NC) electrode, it was used to determine the lead content in a certified reference material of a wastewater sample (SPS-WW1 Batch 113). The collected wastewater sample was diluted with water, distilled in a 1:1 ratio, alkalized with NaOH to a pH of about 5.5, and then analyzed using the proposed electrode. The measurements were repeated three times. The lead content obtained was 98.3 ± 3.8 µg L^−1^/L, which was consistent with the certified value (100 ± 0.5 µg L^−1^), which confirms the correct operation of the developed electrode and its usefulness in the analysis of real samples.

## 4. Conclusions

A new material— a nanocomposite of carbon nanofibers and 1-hexyl-3-methylimidazolium hexafluorophosphate—is proposed as membrane modifier of a lead ion-selective electrode. This CNFs-HMImPF_6_ nanocomposite performs many functions in the electrode, favorably changing its properties. The modification of membrane through the use of the nanocomposite increases its extraction capacity in relation to lead, thanks to which the obtained electrodes show a lower detection limit and a wider measuring range, as well as more favorable selectivity coefficient values. The nanocomposite has a strongly hydrophobic character and also increases the hydrophobicity of the membrane. Thanks to this, the leaching of active ingredients from the membrane phase is significantly reduced, which results in very good long-term stability of the electrodes. Moreover, modified membranes are characterized by reduced resistance and increased capacitance. Due to this, obtained electrodes showed short response times as well as a stable and reversible potential. The use of the nanocomposite as a component of the ion-sensitive membrane is simple, thanks to which the preparation of the electrode is easier and faster compared to electrodes with an intermediate layer, e.g., conductive polymers or carbon nanomaterials. This is an additional advantage of using a nanocomposite.

## Figures and Tables

**Figure 1 materials-16-01003-f001:**
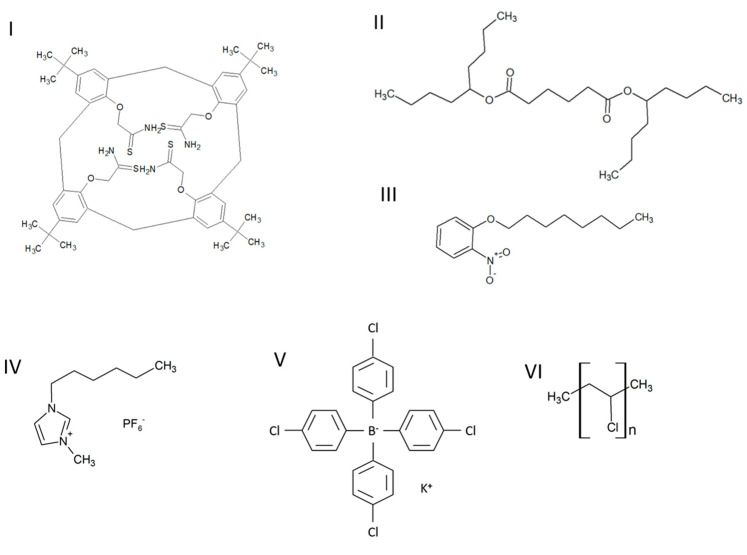
The chemical formulas of the compounds used to prepare the membranes: (**I**) lead ionophore IV, (**II**) bis(1-butylpentyl) adipate, (**III**) 2-nitrophenyl octyl ether, (**IV**) 1-hexyl-3-methylimidazolium hexafluorophosphate, (**V**) potassium tetrakis (p-chlorophenyl) borate and (**VI**) poly(vinyl chloride).

**Figure 2 materials-16-01003-f002:**
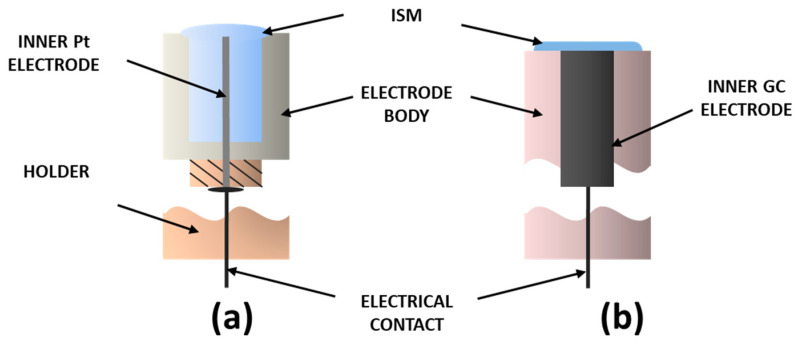
Scheme of the construction of the designed electrodes: (**a**) based on Pt electrode and (**b**) based on GC electrode.

**Figure 3 materials-16-01003-f003:**
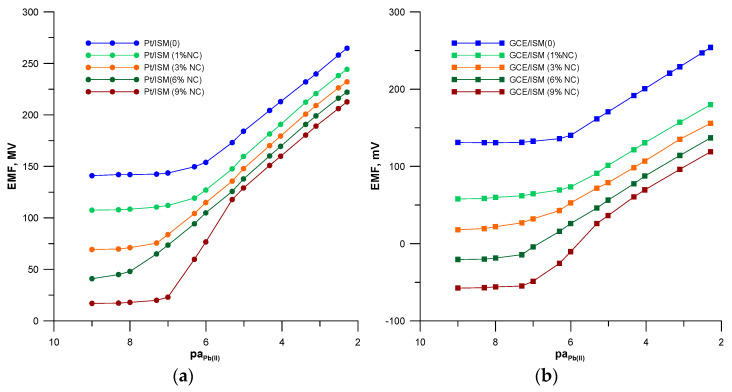
Calibration curves of the studied ISEs: (**a**) based on Pt electrode and (**b**) based on GC electrode.

**Figure 4 materials-16-01003-f004:**
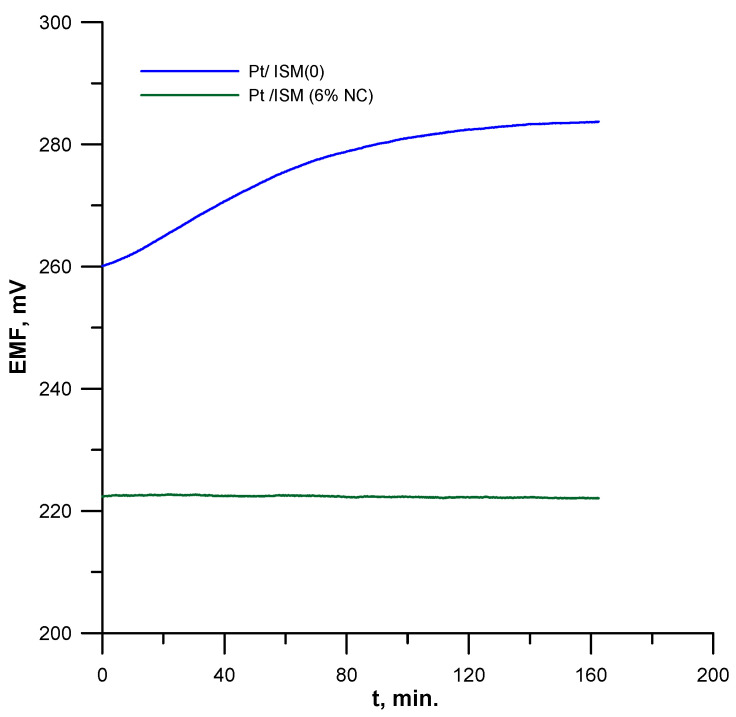
Short-term potential stability of the electrodes under zero current conditions.

**Figure 5 materials-16-01003-f005:**
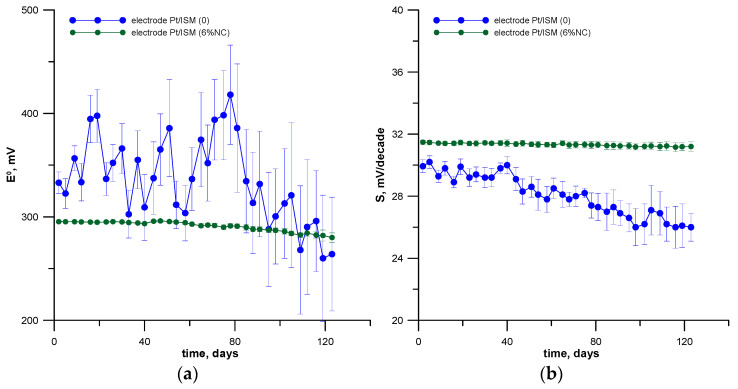
Change of electrode parameters over time for the unmodified electrode (Pt/ISM (0)) and the modified one with the nanocomposite (Pt/ISM (6%NC)): (**a**) E^0^ and (**b**) slope of linear section of response function. Standard deviations given on the plots are determined for the same three electrodes of the two electrode types, i.e., Pt/ISM (0) and Pt/ISM (6%NC), respectively.

**Figure 6 materials-16-01003-f006:**
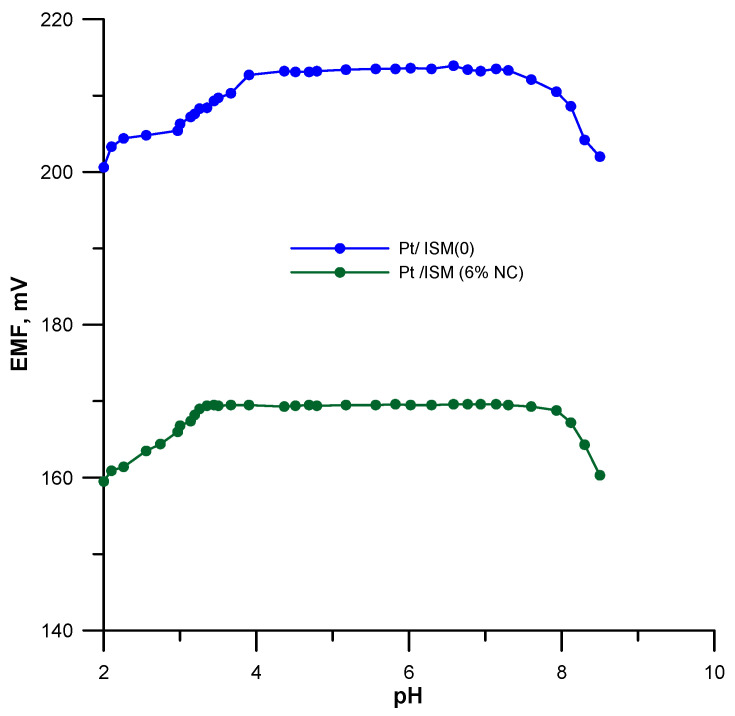
Effect of pH on electrode potential.

**Figure 7 materials-16-01003-f007:**
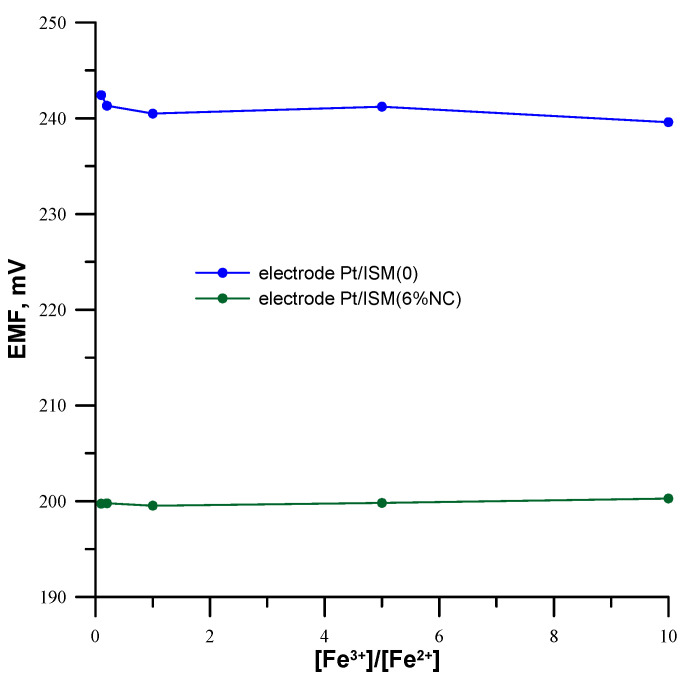
Influence of the redox potential of the sample on the potential of the tested electrodes.

**Figure 8 materials-16-01003-f008:**
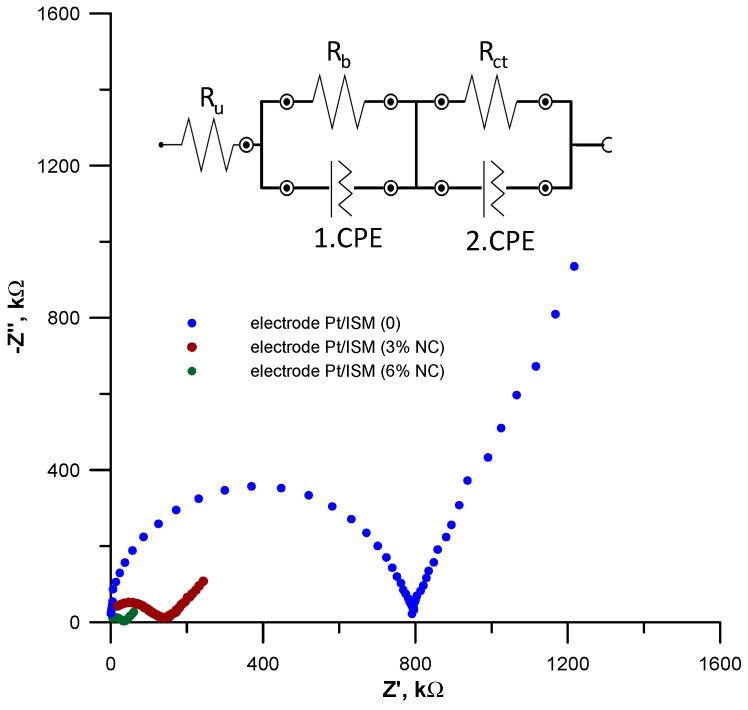
EIS spectra obtained for Pt/ISM (0), (Pt/ISM (3%NC)) and (Pt/ISM (6%NC)) electrodes determined in a 1 × 10^−2^ mol L^−1^ Pb(NO_3_)_2_ solution. The inset shows the equivalent electrical circuit used to fit the data.

**Table 1 materials-16-01003-t001:** Types of tested electrodes and the composition of their membrane.

ISE	Inner Electrode	Membrane Composition, % wt.					
		Ionophore	KTpClPB	NC	PVC	BBPA	NPOE
Pt/ISM (0)	Pt	1	0.5	-	33	33	32.5
Pt/ISM (1%NC)	Pt	1	-	1	33	32.5	32.5
Pt/ISM (3%NC)	Pt	1	-	3	33	32.5	30.5
Pt/ISM (6%NC)	Pt	1	-	6	33	32.5	27.5
Pt/ISM (9%NC)	Pt	1	-	9	33	32.5	24.5
GCE/ISM (0)	GCE	1	0.5	-	33	33	32.5
GCE/ISM (1%NC)	GCE	1	-	1	33	32.5	32.5
GCE/ISM (3%NC)	GCE	1	-	3	33	32.5	30.5
GCE/ISM (6%NC)	GCE	1	-	6	33	32.5	27.5
GCE/ISM (9%NC)	GCE	1	-	9	33	32.5	24.5

**Table 2 materials-16-01003-t002:** Analytical parameters of the studied Pb(II)-ISEs.

Electrode	Slope ^1^, mV/pa	Limit of Detection, mol L^−1^	Linear Range, mol L^−1^	Response Time, s
Pt/ISM (0)	29.9	4.0 × 10^−7^	1 × 10^−6^–1 × 10^−2^	15
Pt/ISM (1%NC)	31.9	2.4 × 10^−7^	5 × 10^−6^–1 × 10^−2^	8
Pt/ISM (3%NC)	31.8	3.1 × 10^−8^	1 × 10^−7^–1 × 10^−2^	5
Pt/ISM (6%NC)	31.5	6.0 × 10^−9^	1 × 10^−8^–1 × 10^−2^	5
Pt/ISM (9%NC)	30.7	3.1 × 10^−6^	5 × 10^−6^–1 × 10^−2^	5
	58.4	-	1 × 10^−7^–5 × 10^−6^	-
GCE/ISM (0)	30.6	5.0 × 10^−7^	1 × 10^−6^–1 × 10^−2^	20
GCE/ISM (1%NC)	28.9	5.1 × 10^−7^	1 × 10^−6^–1 × 10^−2^	12
GCE/ISM (3%NC)	28.2	1.0 × 10^−7^	5 × 10^−7^–1 × 10^−2^	5
GCE/ISM (6%NC)	30.4	2.4 × 10^−8^	5 × 10^−8^–1 × 10^−2^	5
GCE/ISM (9%NC)	29.9	2.8 × 10^−6^	5 × 10^−6^–1 × 10^−2^	5
	52.4	-	5 × 10^−7^–5 × 10^−6^	

^1^ slopes were determined within the linear range given in the fourth column.

**Table 3 materials-16-01003-t003:** Selectivity coefficients values determined for the Pt/ISM (0) and Pt/ISM (6%NC) electrodes.

Interfering Ion	−logK^pot^_Pb(II)/M_	
	Electrode Pt/ISM (0)	Electrode Pt/ISM (6%NC)
Na^+^	4.1	8.3
K^+^	4.6	8.5
Li^+^	4.8	7.2
Ca^2+^	4.9	7.2
Mg^2+^	5.2	8.1
Ba^2+^	4.8	6.4
Cu^2+^	3.0	4.4
Cd^2+^	3.2	5.5
Co^2+^	6.0	6.4
Ni^2+^	6.0	6.5
Zn^2+^	5.8	6.1

**Table 4 materials-16-01003-t004:** Electrical parameters of the studied electrodes determined using equivalent circuits shown in Figure 8 (R_u_-uncompensated series resistance; R_b_ bulk resistance; R_ct_ charge transfer resistance; CPE constant phase element; Y^0^ initial value for the admittance for the CPE element; N- parameter showing to what extent the CPE is the ideal capacitance, if N = 1 then the CPE is the ideal capacitance).

Electrode	R_u_, kΩ	R_b_, kΩ	CPE_1_ Y^0^(N), pF	R_ct_ kΩ	CPE_2_ Y^0^ (N), µF
Pt/ISM (0)	16.8	762	15.4 (0.982)	9832	0.044 (0.991)
Pt/ISM (3%NC)	12.4	178	387 (0.847)	196	20.7 (0.879)
Pt/ISM (6%NC)	11.6	46.5	1231 (0.789)	28.6	82.4 (0.886)

## Data Availability

Not applicable.

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
