# Peer review of "Improved Lead Sensing Using a Solid-Contact Ion-Selective Electrode with Polymeric Membrane Modified with Carbon Nanofibers and Ionic Liquid Nanocomposite"

_materials, 2023, doi:10.3390/ma16031003_

Round 1

Reviewer 1 Report

The authors investigated a new nanocomposite made of carbon nanofibers and an ionic liquid as a solid contact material for lead sensing applications.  However, the manuscript may be suitable for publishing after a major revision.

The following suggestions have been made to improve the manuscript:

 1- The manuscript should be thoroughly reviewed for grammatical, typographical, and punctuation issues.

2- The introduction needs to address a comparison with literature studies in the field of application since the literature survey is rather limited, and indicating the preference of the current work is also required.

3- The abstract needs to be revised to reflect the most significant findings from this study more accurately.

4- There is a need for improved identification of keywords.

5- Including the chemical formulas of the compounds utilized in fabricating electrodes is necessary, in addition to an illustrated diagram of the designed electrodes and their components.

6- The used components of the prepared composite mixtures seem confusing. Therefore, it is better to include the mentioned mixtures in the table.

7- It seems that the results and discussion section lacks scientific discussion and careful analysis of the results obtained, so achieving this point is necessary. Moreover, according to the standards, the figures should be drawn with better quality.

8- The conclusion needs to be completely rewritten, as it appears in its current form as part of an abstract rather than a conclusion.

9- Finally, what are the strategies/recommendations to reduce uncertainties in this study?

Author Response

  • The manuscript should be thoroughly reviewed for grammatical, typographical, and punctuation issues.

In line with the Reviewer’s comment, the entire article was re-checked and previously unnoticed errors were corrected.

  • The introduction needs to address a comparison with literature studies in the field of application since the literature survey is rather limited, and indicating the preference of the current work is also required.

In line with the Reviewer’s comment, the introduction was improved and the use of various nanocomposites in solid contact ion-selective electrodes was discussed.

  • The abstract needs to be revised to reflect the most significant findings from this study more accurately.

According to Reviewer’s comment the abstract is corrected .

  • There is a need for improved identification of keywords.

In line with the Reviewer’s comment, keywords were corrected to improve their identification.

  • Including the chemical formulas of the compounds utilized in fabricating electrodes is necessary, in addition to an illustrated diagram of the designed electrodes and their components.

In line with the Reviewer’s comment, we have added two new Figures 1 and 2. Figure 1 shows the chemical formulas of the compounds used to prepare the membranes, while Figure 2 presents scheme of the construction of the designed electrodes.

  • The used components of the prepared composite mixtures seem confusing. Therefore, it is better to include the mentioned mixtures in the table.

In line with the Reviewer’s comment, we have added new Table 1 in which types of tested electrodes and the composition of their membrane is presented.

  • It seems that the results and discussion section lacks scientific discussion and careful analysis of the results obtained, so achieving this point is necessary. Moreover, according to the standards, the figures should be drawn with better quality.

In line with the Reviewer’s comment, analysis of the obtained results were done and additional explanation were added in appropriate subsections of part 3. Results and discussion. All figures have been corrected.

  • The conclusion needs to be completely rewritten, as it appears in its current form as part of an abstract rather than a conclusion.

In line with the Reviewer’s comment, the conclusions were rewritten.

  • Finally, what are the strategies/recommendations to reduce uncertainties in this study?

We have made every effort to minimize the uncertainty in our research.

First of all, all activities were performed very carefully using good analytical practices.

Measurements were made using instruments characterized by the highest possible measurement accuracy. For each type of electrode, 3 identical sensors were made and tested in parallel.

Each measurement was repeated at least three times.

We hope our manuscript is now suitable for publication in Materials.

                                                            Sincerely,

                                                            Malgorzata Grabarczyk

Reviewer 2 Report

The MS entitled " Improved lead sensing using solid contact ion-selective electrode with polymeric membrane modified with carbon nanofibers and ionic liquid nanocomposite" by Cecylia Wardak et al. aimed the development of a new solid contact ion-selective electrode (ISE) sensitive to lead(II) ions obtained by modifying a polymer membrane with a nanocomposite of carbon nanofibers and an ionic liquid 1-hexyl-3-methylimidazolium hexafluorophosphate. However, I do not find the work new or original as there is already a paper in literature on the same experiments, only the carbon nanofibers are missing, also written by the same author, and a conference paper. Moreover, there is another paper using the same polymer and MWCNT for lead determination. In the light of the above I do not recommend the MS publication in the highly ranked Materials journal. 

Author Response

Thank you very much for taking the time to review our work.

The MS entitled " Improved lead sensing using solid contact ion-selective electrode with polymeric membrane modified with carbon nanofibers and ionic liquid nanocomposite" by Cecylia Wardak et al. aimed the development of a new solid contact ion-selective electrode (ISE) sensitive to lead(II) ions obtained by modifying a polymer membrane with a nanocomposite of carbon nanofibers and an ionic liquid 1-hexyl-3-methylimidazolium hexafluorophosphate. However, I do not find the work new or original as there is already a paper in literature on the same experiments, only the carbon nanofibers are missing, also written by the same author, and a conference paper. Moreover, there is another paper using the same polymer and MWCNT for lead determination. In the light of the above I do not recommend the MS publication in the highly ranked Materials journal. 

The paper describes the use of a new material (nancomposite of 1-hexyl-3-methylimidazolium hexafluorophosphate and carbon nanofibers) that performs many functions in the electrode, favorably changing its properties. By using only one other component of the membrane (compared to conventional composition) many advantages were achieved. These include lowering the membrane resistance and reducing the charge transfer resistance at the membrane / inner electrode interface, which in turn shortens the response time and significantly improves the stability and reversibility of the electrode potential. In addition, the modification of the membrane with the addition of a nanocomposite increases its extraction capacity in relation to lead, thanks to which the obtained electrodes show a lower detection limit and a wider measuring range as well as more favorable values of selectivity coefficients. The nanocomposite has a strongly hydrophobic character and also increases the hydrophobicity of the membrane. Thanks to this, the leaching of active ingredients from the membrane phase is significantly reduced, which allows for very good long-term stability and extends the lifetime of the electrodes.

The use of the nanocomposite as a component of the ion-sensitive membrane is simple, thanks to which the preparation of the electrode is easier and faster compared to electrodes with an intermediate layer, e.g. conductive polymers or carbon nanomaterials. This is an additional advantage of the electrode developed by us.

Mentioned in this review, our earlier work concerns the use of only ionic liquid as a component of a polymer membrane. In the construction presented there, it was necessary to use an intermediate layer of conductive material (in this case it was poly(3-octyl)thiophene (POT)). The results of these measurements were the starting point for us to undertake research on the development of new electrodes using the nanocomposite. Electrodes with a nanocomposite in the membrane, compared to electrodes containing only ionic liquid in the membrane, show much better long-term stability and their preparation is faster and simpler.

We hope that we have convinced the Reviewer of the novelty of our manuscript and now it is suitable for publication in Materials.

Sincerely,

                                                                           Malgorzata Grabarczyk

Reviewer 3 Report

The present paper is describing new electrodes based on ionic liquids and carbon nanofillers for the detection of lead ions. This topic is in a high demand for a scientific community and achived results are interesting. However, some important experiments are missing and the presentation quality is not on a high level. more detailed comments are listed below:

1) Authors should explain why they use this ioni liquid (how they choose counter anion and cation)

2) line 22-24, please give values which prove that this electrode was better (tange of linear regime? rsponse time? something else?)

3) Part about lead in introduction (lines 32-47) could be shortened

4) line 48, please leave only one word "better", thus it would be more suitable for scientific style

5) Please, provide refs for statement on lines 72-74

6) part in lines 82-84 should be explained in more details (while parts about toxic effects of lead and potentiometry could be shortened for 2-3 sentences). Please, describe prevuoisly reported studies and the role of ionic liquids in them. Cite some key works ( 10.3390/molecules25245812, 10.1016/j.jpowsour.2018.04.053, 10.1002/pola.28937)

7) line 106, describe CNFs (length and diameter) based on supplier sourse or add respective characterization in other section

8) section 2.3, add temperature at which measurements were performed

9) check the order of figures and tables - they should appear after first mention in the text and not as one by one

10) please revise table 1. It's too hard to understand which line respect to which sample

11) in section 3.3 please add information about literature data of coefficient selectivity

12) section 3.6 - add information about tempearture at which measurements were performed. Additional experiments about condictivity of other electrode compositions (at least 1 more) are required

13) please provide data about inetractions of ILs with carbon fillers. For examples, FTIR spectra

14) what is temperature effect on the measurements? In what temperature range detection could be done?

Author Response

Thank you very much for the valuable comments.

The present paper is describing new electrodes based on ionic liquids and carbon nanofillers for the detection of lead ions. This topic is in a high demand for a scientific community and achived results are interesting. However, some important experiments are missing and the presentation quality is not on a high level. more detailed comments are listed below:

1) Authors should explain why they use this ioni liquid (how they choose counter anion and cation)

The assumption of our research was to modify the ion-selective membrane by using a nanocomposite that will simultaneously perform several functions in the obtained electrode.

 We were looking for a substance that creates a composite with carbon nanofibers and at the same time will positively affect the extraction capacity of the membrane in relation to lead ions.

Such conditions are met by the ionic liquid 1-hexyl-3-methylimidazolium hexafluorophosphate  that we have chosen. In line with the Reviewer’s comment, explanations regarding the choice of this ionic liquid have been added in the introduction of the manuscript.

2) line 22-24, please give values which prove that this electrode was better (tange of linear regime? response time? something else?)

In line with the Reviewer’s advice, additional data regarding analytical parameters of proposed electrode was provided in the abstract.  

3) Part about lead in introduction (lines 32-47) could be shortened

In line with the Reviewer’s advice, the introduction was corrected and part about lead is shortened in the revised manuscript.

4) line 48, please leave only one word "better", thus it would be more suitable for scientific style

In line with the Reviewer’s comment, appropriate correction was made.

5) Please, provide refs for statement on lines 72-74

In line with the Reviewer’s comment, new citation were added. They are cited as refs [14, 15] in the revised manuscript.

6) part in lines 82-84 should be explained in more details (while parts about toxic effects of lead and potentiometry could be shortened for 2-3 sentences). Please, describe prevuoisly reported studies and the role of ionic liquids in them. Cite some key works ( 10.3390/molecules25245812, 10.1016/j.jpowsour.2018.04.053, 10.1002/pola.28937)

In line with the Reviewer’s comment, introduction was corrected. It was shortened and the previously reported studies with the application of ionic liquid are described and relevant literature cited.

7) line 106, describe CNFs (length and diameter) based on supplier sourse or add respective characterization in other section

In line with the Reviewer’s comment, data on the length and diameter of the CNFs have been added in the Reagents section.

8) section 2.3, add temperature at which measurements were performed

All measurements were made at room temperature (25°C±1). This information was added ion experimental section.  

9) check the order of figures and tables - they should appear after first mention in the text and not as one by one

In accordance with the Reviewer's comment, the order of figures and tables has been changed, placing them after the first mention in the text.

10) please revise table 1. It's too hard to understand which line respect to which sample

In line with the Reviewer’s comment, Table 1 which is Table2 in the revised manuscript was corrected.   

11) in section 3.3 please add information about literature data of coefficient selectivity

In line with the Reviewer’s comment, relevant literature cited as ref [40].

12) section 3.6 - add information about tempearture at which measurements were performed. Additional experiments about condictivity of other electrode compositions (at least 1 more) are required

In line with the Reviewer’s comment, the information about temperature at which measurements were performed is added in experimental part. Moreover EIS results obtained for Pt/ISM(3%NC) electrode are added in Figure 8 and in Table 4.

13) please provide data about inetractions of ILs with carbon fillers. For examples, FTIR spectra

Unfortunately, we are unable to perform additional FTIR measurements in such a short time.

Our statement regarding the interaction of ionic liquids with carbon nanofibers was supported by relevant examples described in the literature (ref. [38,39]).

14) what is temperature effect on the measurements? In what temperature range detection could be done?

Unfortunately, we did not study in detail the effect of temperature on the values of individual analytical parameters of the electrodes presented in this work. This is a very interesting suggestion and we will do such research in the future.

Based on our experience so far, we can say that, with increasing temperature, the slope of the electrode characteristic will increase. We can also conclude that the optimal measurement temperature is 20 - 25 °C.

We hope our manuscript is now suitable for publication in Materials.

                                                            Sincerely,

                                                            Malgorzata Grabarczyk

Round 2

Reviewer 1 Report

The response to all the comments by the authors is appreciated, and the revised manuscript version now appears in its best form, so the recommendation is to publish the manuscript.

Reviewer 3 Report

Authors addressed all questions. This article could be published in the present form